# Trajectory analyses in insurance medicine studies: Examples and key methodological aspects and pitfalls

**Laura Serra**[1,2,3]*, **Kristin Farrants**[4], **Kristina Alexanderson**[4], **Mónica Ubalde**[1,5], **Tea Lallukka**[6]

**1** Center for Research in Occupational Health (CiSAL), University Pompeu Fabra, Barcelona, Spain, **2** Research Group on Statistics, Econometrics and Health (GRECS) University of Girona, Girona, Spain, **3** CIBER of Epidemiology and Public Health (CIBERESP), Madrid, Spain, **4** Division of Insurance Medicine, Department of Clinical Neuroscience, Karolinska Institutet, Stockholm, Sweden, **5** Barcelona Institute for Global Health (ISGlobal), University Pompeu Fabra, Barcelona, Spain, **6** Department of Public Health, University of Helsinki, Helsinki, Finland

* laura.serra-saurina@upf.edu

## Abstract

### Background

Trajectory analyses are being increasingly used in efforts to increase understanding about the heterogeneity in the development of different longitudinal outcomes such as sickness absence, use of medication, income, or other time varying outcomes. However, several methodological and interpretational challenges are related to using trajectory analyses. This methodological study aimed to compare results using two different types of software to identify trajectories and to discuss methodological aspects related to them and the interpretation of the results.

### Methods

Group-based trajectory models (GBTM) and latent class growth models (LCGM) were fitted, using SAS and Mplus, respectively. The data for the examples were derived from a representative sample of Spanish workers in Catalonia, covered by the social security system (n = 166,192). Repeatedly measured sickness absence spells per trimester (n = 96,453) were from the Catalan Institute of Medical Evaluations. The analyses were stratified by sex and two birth cohorts (1949–1969 and 1970–1990).

### Results

Neither of the software were superior to the other. Four groups were the optimal number of groups in both software, however, we detected differences in the starting values and shapes of the trajectories between the two software used, which allow for different conclusions when they are applied. We cover questions related to model fit, selecting the optimal number of trajectory groups, investigating covariates, how to interpret the results, and what are the key pitfalls and strengths of using these person-oriented methods.

**Data Availability Statement:** The data that support the findings of this study come, on the one hand, from an annual sample of Spanish Social Security affiliates that are part of the Continuous Working

Life Sample (MCVL by its acronym in Spanish), provided by the General Directorate of Social Security (DGOSS by its acronym in Spanish) and, on the other hand, from the linked records of sickness absence episodes certified in Catalonia, provided by Catalan Institute for Medical and Health Evaluations (ICAM by its acronym in Catalan). Future researchers should contact these two public institutions in order to request access to the data underlying this study as they are not publicly available. To access those sensitive data, as in most countries, ethical permission from the researcher's institution and the respective data holding authorities is needed. All researchers granted such permissions can access these type of data. The MCVL data can be obtained filling the PDF form available online at https://www.seg-social.es/wps/portal/wss/internet/EstadisticasPresupuestosEstudios/Estadisticas/EST211/1459 and sending it to the DGOSS (mcvl.dgoss-sscc@seg-social.es) after having acquired ethical permission for handling these type of sensitive data. These data files are generated annually and DVDs with older versions of the MCVL can be requested. The versions from 2012 to 2014 were used for this study. Access to linked records of sickness absence episodes provided by the ICAM was possible thanks to an agreement signed between DGOSS, ICAM and CiSAL-UPF. This records linkage is made under strict confidentiality control and is done directly by DGOSS and ICAM. The resulting pseudonymised data can be requested by contacting the ICAM (bustia.icam@gencat.cat) and reaching an agreement to obtain the sickness absence data (ITCC by its acronym in Catalan) from 2012 to 2014. Authors of this study had no special access privileges to the data and other researchers would obtain the same information after being granted their access by the DGOSS and the ICAM.

**Funding:** TL is supported by the Academy of Finland (Grant #330527) and the Social Insurance Institution of Finland (grant 29/26/2020).

**Competing interests:** The authors have declared that no competing interests exist.

## Conclusions

Future studies could address further methodological aspects around these statistical techniques, to facilitate epidemiological and other research dealing with longitudinal study designs.

## Introduction

Results from occupational epidemiology, insurance medicine, and social security schemes research are commonly building on assessing exposure-outcome association measures at one time point, or using dichotomous outcomes such as modelling time to an event (yes or no). To better understand the associations between health and work there is a need of a more longitudinal perspective that applies methods to identifying *time*-varying effects in *association measures. Thus, there is a need for a methodological paper in this area.*

In recent years, improvements in data collection have allowed new methodological approaches to analyse time-varying data in longitudinal study designs in occupational epidemiology [1, 2]. Administrative registers usually provide an excellent possibility to make use of a large amount of longitudinal data, which is useful when applying time-related statistical techniques [3]. In particular, trajectory analysis approach is increasingly applied in longitudinal studies in occupational epidemiology, social security, and public health [4]. This methodology allows reconstructing the course of exposures and outcomes over time, to identify patterns and detect specific groups following similar development in the outcome in the studied cohorts. This has led to better understanding of the existence of different patterns within populations over time, than the traditional methods of studying changes in population averages. Trajectory analyses have been used to identify, e.g., different development over time in the number of days on sickness absence and/or disability pension, unemployment days, as well as of medication use, cost of sick leave or of costs of illness, over different time spans, typically during a number of months or years [5–11].

In particular, latent growth modelling approaches, such as latent class growth analysis (LCGA) and growth mixture modelling (GMM) have been increasingly applied for their usefulness to summarise huge amounts of information and identify homogeneous subpopulations which are easier to handle and interpret [12].

We use the definition of group-based trajectory modelling (GBTM) as proposed by Nagin and Ogden: "finite mixture modelling application that uses trajectory groups as a statistical device for approximating unknown trajectories across population members" [13]. The GBTM definition differs from GMM, as it does not assume that there are distinct subpopulations within the population, but rather uses the trajectory groups as a statistical tool to approximate the heterogeneity within a population [13]. LCGA and GBTM are sometimes treated as synonymous [14], however, others consider them as different, and consider GBTM a special case of LCGA where the error variance is assumed to be the same for all classes and all time points [15]. To what extent the two methods are comparable is therefore unclear and often discussed among researchers.

Both GBTM and LCGA assume that the variation in individual trajectories can be summarised with a finite number of polynomial functions [12, 16]. Nagin, - _blankNagin, - _blankThey also assume that the trajectory within each group is homogenous [12, 17]. Finally, they assume conditional independence, whereby the current value is assumed to be independent of past values [16]. However, this assumption is made at the group level, not the

individual level–an individual's current value is assumed to be conditional on the trajectory group [17].

Therefore, many methodological and interpretative questions arise now, when we use to a larger extent use trajectory analyses of longitudinal data. First, there is a debate about the potential usefulness of different software tools available (i.e., SAS, Mplus, Stata, or R), as well as the potential divergences in terms of their efficiency and results [13, 18–20]. Another aspect is the criteria for determining the optimum number and shape of trajectories [21]. Finally, a discussion on how to interpret results, e.g., regarding differences between trajectory groups is warranted [22].

In all, there is a general need in the scientific community to be able to select the most appropriate software and to properly understand and well interpret the results, being aware of all the possible strengths and weaknesses of the data and the selected method. Thus, the aims of this methodological study were, first to compare results using two different types of software in trajectory analyses, and second, to discuss methodological aspects of trajectory analysis highlighting pitfalls/setting out doubts and suggesting solutions to situations that might arise when deciding the statistical strategy, interpreting outputs and implications of results.

## Methods

### Methodological approach to compare available software to deal with trajectory analysis

The advances and availability of statistical analysis software designed for trajectory analyses give rise to discussions about which of them would be the most appropriate option in operative and computational terms (depending on the available data, the outcome, and research question), and how much results could change due to the differences between different models/procedures in identifying trajectories.

Within the growth modelling framework, we can distinguish between three types of models: multilevel and mixed linear models, which are mostly the same, and the structural equation modelling (SEM) which differs from the other two in terms of (1) treatment of time scores and (2) treatment of time-varying outcomes (or covariates).

In this study, we compared results from group-based trajectory models (GBTM) and latent class growth models (LCGM). Even if there are other different types of software, this study focused on SAS, which is based on group-based models that estimate a discrete mixture model for clustering of longitudinal data series, and on Mplus (version 7.11), which is considered the most powerful software concerning convergence [23]. It is important to point out that our analysis using Mplus was based on Latent Class Growth Analysis (LCGA) instead of Growth Mixture Models (GMM), which is less computationally demanding, does not lead to convergence issues, and requires smaller samples [24]. In addition, SAS only deals with LCGA so another reason to use this method was to allow more accurate comparisons between results derived using different software.

Thus, the aim of this first part was to compare and discuss results from both software (Mplus and SAS) considering a longitudinal study from the same dataset regarding operative and computational aspects.

The data used were derived from a representative sample of workers covered by the social security system and living in Catalonia, Spain (n = 166,192) [25]. Information from registered sickness absence spells (96,453 spells and n = 166,192 workers) come from the Catalan Institute of Medical Evaluations. For the trajectory analyses, the number of days on sickness absence per trimester was used as a repeated measure, stratified by sex and two birth cohorts (1949–1969 and 1970–1990).

To better interpret the results, it is important to note that this is a time-structured study, where all individuals are assessed at exactly the same time intervals. In addition, all individual growth trajectories within a class are assumed to be homogeneous, and the variance and covariance estimates for the growth factors within each class are assumed to be fixed to zero.

In theoretical terms, certain differences between the two software are known. Thus, SAS (PROC TRAJ) is based on group-based models (GBTM) while Mplus is based on SEM. In SAS, missing is assumed to be completely at random, and a complete case analysis is a default setting of the model. SAS can handle missing data in some of the measurement points, but if missing data are due to previous time point(s) (attrition is selective [26]), a drop-out model is recommended. It is also possible to perform imputation (PROC MI to create the imputed data-sets & PROC MINALYZE for analysing the imputed data). For time-variant covariates, if used, missing data result in participants being excluded from the analyses. If there are missing data, and dropout model is not used, group sizes will be biased, but trajectory shapes are not affected. In Mplus imputation or considering missing data are not required, as Mplus can estimate a model with missing values. Then, even if both Mplus and SAS incorporate user flexibility in terms of modifying starting values to avoid local solutions, Mplus also offers information about which could be the best value that could be used as the seed value for checking the model parameter estimates. Mplus offers different tests of model fit to help determine the optimal number of trajectories while SAS give less output automatically. Finally, Mplus suggest that results are better when one considers the same number of growth factors in all trajectories.

## Results

### How to interpret the results from trajectory analyses?

There are many choices to be made when conducting trajectory analyses and each choice raises questions of what this means for the interpretation of the results. Questions that arise in the analysis and interpretation process concern how to fit the final model, how many trajectories to choose, how well the trajectories represent actual trajectories of individuals, and how to relate group membership to covariates.

Regarding measures of model fit, several measures can be used. The log likelihood, which gives an indication of model fit without considering number of parameters, is much used. The Bayesian Information Criterion (BIC) and the Akaike Information Criterion (AIC) are, however, most widely used for determining the number of trajectories (mainly the BIC value), since they punish models for each added parameter–this allows a balance between model accuracy and model parsimony [27]. In addition, statistical tests such as the LMR-LRT and the bootstrap likelihood ratio test (BLRT) show a statistically significant difference between the (k)-class versus (k+1)-class models. Finally, researchers also follow other criteria that can help to reach the final decision: e.g., 5% of the participants in each group or an average of the maximum posterior probability of assignments above 70% in all classes [28]. Adequacy of the model can be further assessed using odds of correct classification, mismatch scores (higher than 5) and entropy (values greater than 0.5) [22]. There is also an element of researcher judgement here, since the groups should be meaningfully different from each other and allow for some interpretation of these differences.

In practical terms, only results for women and for the latest cohort (1949–1969) are shown. Thus, in Mplus, according to the significant Lo, Mendell and Rubin likelihood ratio test (LMR-LRT) statistic [21], the optimal number of trajectories was three. To be more specific, three trajectories each with a cubic term (shape of the trajectory) were identified. This result was maintained according to the value of Entropy, which was larger compared to a four

**Table 1. Model fit information from Mplus of various models of trajectories of sickness absence days per quarter (2012–2014) considering data from women of working age, living in Spain, and born in 1949–1969.**

| Number of trajectories | Entropy | BIC | Sample-size adjusted BIC | AIC | BLRT | LMR-LRT | Log likelihood | APPA[1a1] | % classes[2] |
|---|---|---|---|---|---|---|---|---|---|
| 2 full model* (cubic trajectories) | 0.825 | 150,672.5 | 150,605.7 | 150,526.2 | <0.001 | <0.001 | -76,587.1 | 0.86 | 11% (897) |
|  |  |  |  |  |  |  |  | 0.96 | 88.5% (6,909) |
| **3 full model (cubic trajectories)** | **0.715** | **149,682.4** | **149,599.8** | **149,501.4** | **<0.001** | **0.004** | **-75,242.1** | **0.90** | **79.5% (6,209)** |
|  |  |  |  |  |  |  |  | **0.82** | **10.6% (831)** |
|  |  |  |  |  |  |  |  | **0.82** | **9.8% (765)** |
| 3 variable | 0.709 | 149,761.9 | 149,688.8 | 149,601.7 | <0.001 | 0.029 | -75,242.1 | 0.90 | 79.0% (6,164) |
|  |  |  |  |  |  |  |  | 0.81 | 10.8% (847) |
|  |  |  |  |  |  |  |  | 0.81 | 10.2% (795) |
| **4 full model (cubic trajectories)** | **0.695** | **149,057.3** | **148,958.8** | **148,841.5** | **<0.001** | **0.083** | **-74,724.7** | **0.76** | **5.9% (462)** |
|  |  |  |  |  |  |  |  | **0.87** | **75.3% (5,876)** |
|  |  |  |  |  |  |  |  | **0.78** | **8.7% (676)** |
|  |  |  |  |  |  |  |  | **0.82** | **10.1% (792)** |
| 4 variable | 0.662 | 149,085.6 | 148,999.8 | 148,897.6 | <0.001 | <0.001 | -74,813.3 | 0.76 | 9.8% (762) |
|  |  |  |  |  |  |  |  | 0.86 | 73.0% (5,701) |
|  |  |  |  |  |  |  |  | 0.78 | 9.1% (713) |
|  |  |  |  |  |  |  |  | 0.77 | 8.1% (630) |
| 5 full model (cubic trajectories) | 0.740 | 148,990.6 | 148,876.2 | 148740.0 | <0.001 | 0.671 | -74,393.6 | 0.77 | 8.4% (656) |
|  |  |  |  |  |  |  |  | 0.77 | 5.4% (421) |
|  |  |  |  |  |  |  |  | 0.80 | 10.0% (778) |
|  |  |  |  |  |  |  |  | 0.88 | 75.5% (5,893) |
|  |  |  |  |  |  |  |  | 0.81 | 0.7% (57) |

[1]Average Latent Class Probabilities for Most Likely Latent Class Membership by Latent Class.

[2]Based on estimated posterior probabilities

- *Full model: when using the same polynomial order for each growth factor.

trajectory model (solution). On the one hand, based on the BIC values, a four-cubic trajectories solution was suggested to be the best one (Table 1). On the other hand, using SAS and after analysing two to five trajectory group models considering up to the cubic growth factor, the final solution was the one that considered four trajectories (one constant and the others cubic) (Table 2). The decision regarding the number of trajectory groups was based on the BIC value, and the size of the smallest trajectory group.

The next step was to analyse data and plots visually. This gave an indication of the starting value and shape of each trajectory, as well as the 95% confidence intervals (CI) for each trajectory. The decision regarding the polynomial order of each group was based on both the visual inspection of the trajectories plot and on the statistical significance of the growth factors.

The analysis of the CIs of the graphic representations might be a useful tool to know the level of uncertainty in the model. There are, nevertheless, reasons to be cautious when interpreting such CIs, as they can give a misleading idea of the variance, or lack thereof, within each group. As the trajectory groups are only approximations, for some people, even if they are placed in a specific group, their development might differ from that of the other members in that group. It is easy to misinterpret CIs as if the individual trajectory of every individual is within the CI or at least close to either the upper or lower bound. However, CIs do not tell us about the spread of individual trajectories, but rather the level of uncertainty about the trajectory as a whole. Additionally, if the data are small and group size is below 10%, the CIs are likely wider. The most important point is that as the aim of the trajectory modelling is to

**Table 2. Model fit information from SAS[1] of various models of trajectories of sickness absence days per quarter (2012–2014) considering data from women of working age, living in Spain, and born in 1949–1969.**

| Number of trajectories | Entropy | BIC | Sample-size adjusted BIC | AIC | BLRT | LMR-LRT | Log likelihood | APPA[2] | %classes |
|---|---|---|---|---|---|---|---|---|---|
| 2 full model (cubic trajectories) | - | -75,940.8 | -75,936.9 | -75,902.1 | - | - | -75,892.1 | | 88.0% |
| | | | | | | | | | 12.8% |
| 3 full model (cubic trajectories) | - | -75,538.4 | -75,532.6 | -75,480.4 | - | - | -75,465.4 | | 12.4% |
| | | | | | | | | | 79.0% |
| | | | | | | | | | 8.6% |
| 4 full model (cubic trajectories) | - | -75,331.5 | -75,323.8 | -75,254.1 | - | - | -75,234.1 | | 10.7% |
| | | | | | | | | | 76.1% |
| | | | | | | | | | 6.6% |
| | | | | | | | | | 6.6% |
| **4 variable (3 cubic and 1 constant)** | **-** | **-75,320.4** | **-75,313.8** | **-75,254.6** | **-** | **-** | **-75,237.6** | **0.74** | **6.6%** |
| | | | | | | | | **0.87** | **76.0%** |
| | | | | | | | | **0.77** | **10.7%** |
| | | | | | | | | **0.79** | **6.6%** |
| 5 full model (cubic trajectories) | - | -75,355.9 | -75,346.2 | -75,259.1 | - | - | -75,234.1 | | 10.7% |
| | | | | | | | | | 76.0% |
| | | | | | | | | | 0.2% |
| | | | | | | | | | 6.6% |
| | | | | | | | | | 6.6% |

[1]Version 9.4. SAS Institute 2013.

[2]Average Latent Class Probabilities for Most Likely Latent Class Membership by Latent Class.

identify homogeneous groups of people, the CIs are assumed to be very narrow in a reliable model where posterior probabilities are high.

Since CIs do not describe the variability within each group, a way to visualise the variability is through spaghetti plots over individual trajectories within each group. The point is to see for each group, how much variability is there within the group. These graphs allow visualising how much variability there is between individual units at a given time or measuring how much variance there is within units over time or measure. A plot with separate lines for each unit was drawn where the space amongst lines represents the variability between units and the change in each line (slope) represents within-unit variability. Fig 1 (supplementary material) shows the spaghetti plots of individual trajectories within each group. Although the spaghetti plots do vaguely cluster around the general trajectory estimated for the respective groups, they contain far more variation than is suggested by the CIs.

Then, looking at the graphical representation considering four trajectories in both SAS and Mplus (Fig 2), even if we can identify similar trajectories, in the Mplus output there are important differences in the start values of the trajectories. Thus, looking at how individuals were distributed in the four trajectories according to some labour characteristics (type of contract, occupational category and working time), differences were also observed (Table 3).

Therefore, in our example it was observed that growth factors differed somewhat according to the software. Moreover, also the optimal number of trajectories was not clear using one software or the other, as individuals were distributed differently in trajectories and differences in the initial values of the trajectories were detected.

Finally, when dealing with trajectory analysis, many researchers take the approach of performing a multinomial logistic regression over the association between covariates and trajectory group membership and once all model fitting is already conducted. While this is a useful

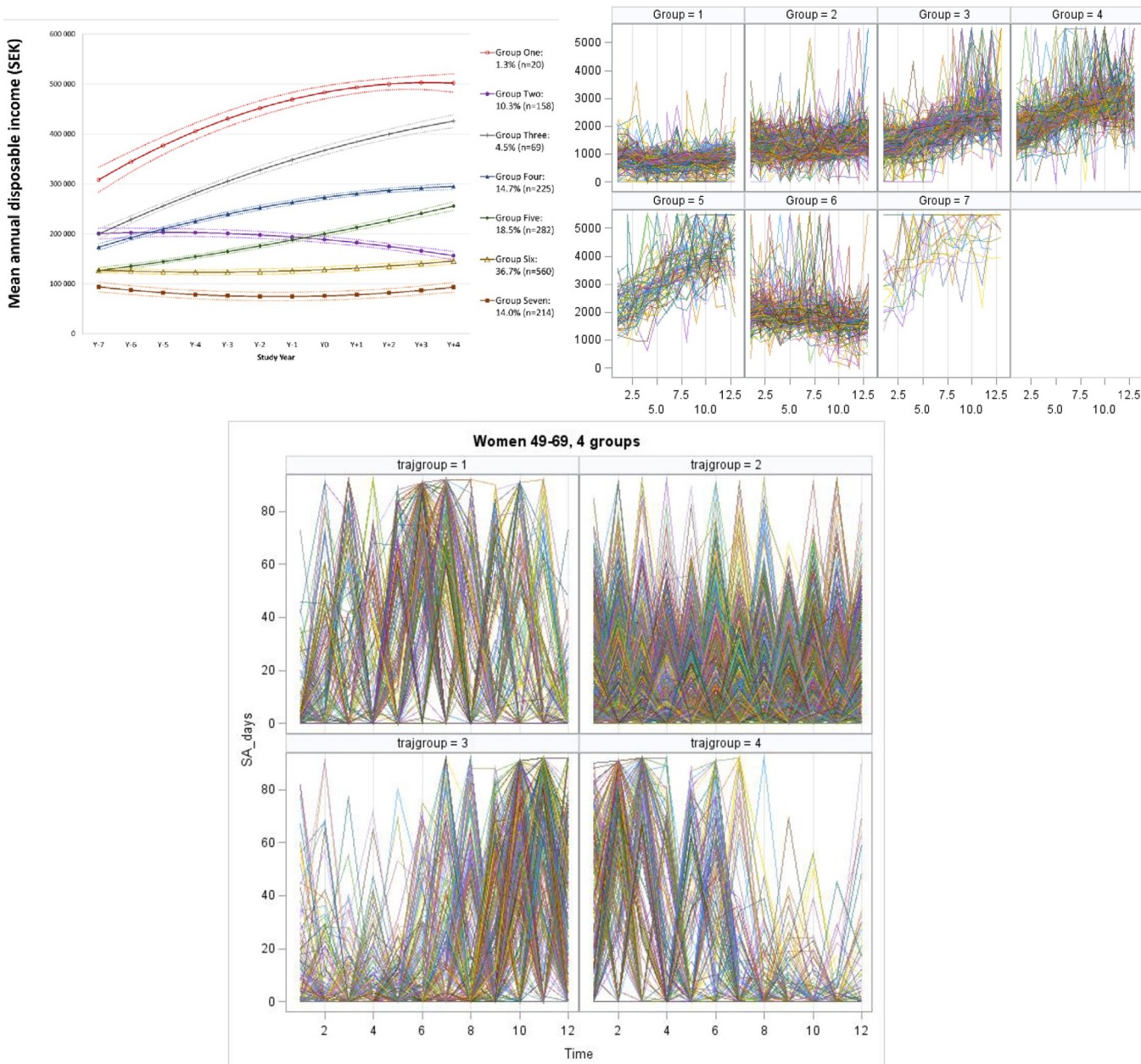

**Fig 1. Spaghetti plots over days in sickness absence per trimester (2012–2014) by trajectory group according to group-based trajectory modelling, among a representative sample of women born in 1949–1969, registered with the social security system, and living in Catalonia, Spain.**

method of analysis, there are certain aspects that need to be considered when performing such analyses. The conversion from probabilities of group membership into most likely group means that a lot of information is lost. Furthermore, logistic regression assumes that the actual group membership for each individual is known, when in reality, it is an estimated probability that has been converted to group membership. There are several ways to deal with this: the first is to add covariates at the same stage as model fitting, although this can complicate the model building substantially [29, 30]. Of the several methods to relate class membership to covariates, the one that has been shown to perform the best is the 3-step approach developed

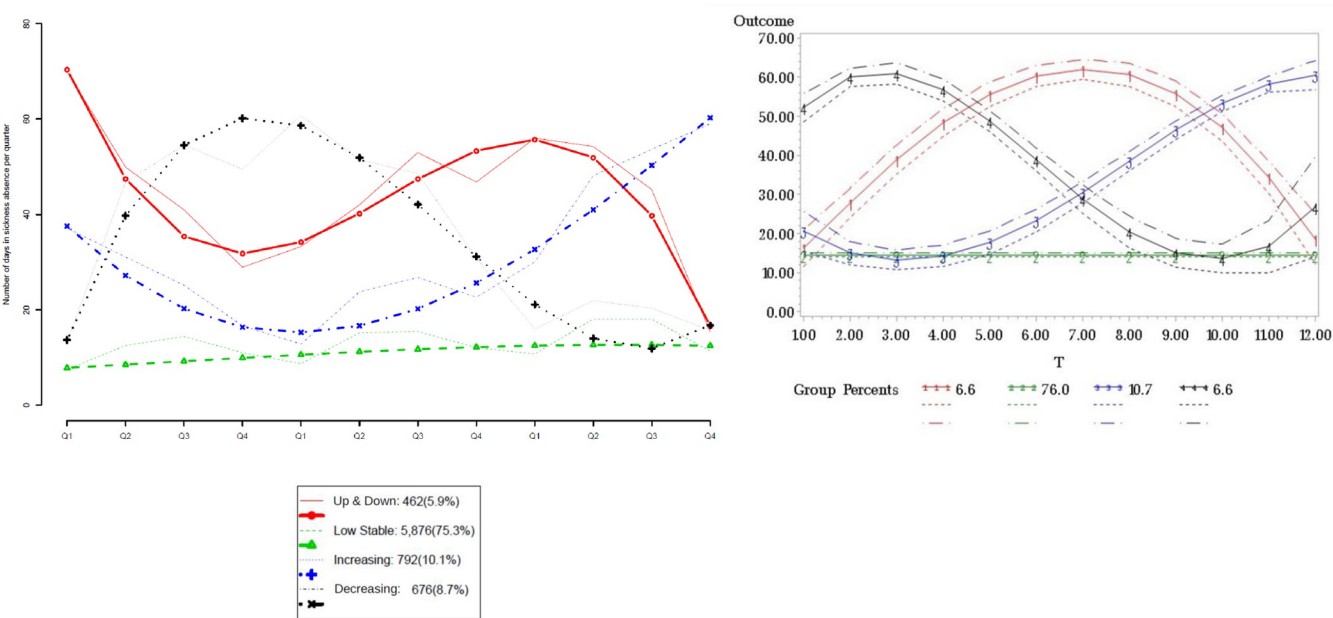

*Note that the X-axis is differently named due to different software. Thus, the graph produced using Mplus is labelled Q1-Q4 for each trimester of 2012, 2013, and 2014, respectively, whereas the graph using SAS is labelled with each respective time point being numbered from 1-12.

**Fig 2.** Graphical representation of the trajectories considering the number of accumulated days in sickness absence (Y-axis) per quarter (2012–2014) (X-axis*) among women from the cohort (1949–1969) using MPlus (left) and SAS (right) statistical software. * Note that the X-axis is differently named due to different software. Thus the graph produced using Mplus is labelled Q1-Q4 for each quarter of 2012, 2013, and 2014, respectively, whereas the graph using SAS is labelled with each respective time point being numbered from 1–12.

by Vermunt, itself an improvement of Block, Croon and Hagenar's 3-step method [29, 30]. This method takes into account misclassification in group membership assignment using a maximum likelihood procedure. For more information see Vermunt [30] and Davies [29].

Sometimes, baseline measures are not appropriate to deal with changes in covariates or to cope with within-group comparison or with the possibility that events alter group membership. Time-varying covariates can be the solution, considering that no missing data are allowed in them, or participants can be excluded from the analyses (in group based trajectory analyses), potentially leading to selection bias.

## Discussion

This study highlights methodological aspects regarding trajectory analysis, both comparing results from the use of two different software on the same data and also by highlighting the main methodological points that are of major concern among researchers dealing with longitudinal studies.

Trajectory analysis can be a strong tool in exploratory analysis to examine heterogeneity, discover new patterns, and connect patterns to covariates. Additionally, groups that might need special attention and/or intervention can be identified. However, interpretation of results is not as easy as it may look. It requires understanding of the different models, of differences between software and their respective limitations, as well as being aware of possible other pitfalls.

In the example shown in this paper, within-class variability around each group was fixed, allowing variability only between classes. This means that we cannot discuss differences within classes. This aspect makes the two software comparable at the level of this assumption. One

**Table 3. Distribution of individuals in the four trajectories according to three labour characteristics (type of contract, occupational category, and working time).** In the left side results from the Mplus. In the right side, results from SAS.

| Covariates | Up &Down (5.9%) | | Low Stable (75.3%) | | Increasing (10.1%) | | Decreasing (8.7%) | |
|---|---|---|---|---|---|---|---|---|
| | N | % | N | % | N | % | N | % |
| Type of contract | | | | | | | | |
| Permanent | 297 | 87.6 | 5563 | 83.9 | 364 | 85.2 | 340 | 83.5 |
| Temporary | 42 | 12.4 | 1070 | 16.1 | 63 | 14.8 | 67 | 16.5 |
| Occupational category | | | | | | | | |
| Skilled Non-manual | 61 | 18.0 | 1401 | 21.1 | 85 | 19.9 | 73 | 17.9 |
| Skilled Manual | 59 | 17.4 | 1090 | 16.4 | 65 | 15.2 | 56 | 13.8 |
| Unskilled Non-manual | 154 | 45.4 | 3038 | 45.8 | 188 | 44.0 | 193 | 47.4 |
| Unskilled Manual | 65 | 19.2 | 1104 | 16.6 | 89 | 20.8 | 85 | 20.9 |
| Working time | | | | | | | | |
| Full-time | 260 | 76.7 | 4791 | 72.2 | 316 | 74.0 | 298 | 73.2 |
| Part-time | 79 | 23.3 | 1842 | 27.8 | 111 | 26.0 | 109 | 26.8 |
| **Up &Down (6.6%)** | | | **Low Stable (76.0%)** | | **Increasing (10.7%)** | | **Decreasing** | |
| | | | | | | | **(6.6%)** | |
| Covariates | N | % | N | % | N | % | N | % |
| Type of contract | | | | | | | | |
| Permanent | 303 | 85.8 | 5596 | 84.0 | 391 | 84.3 | 274 | 84.6 |
| Temporary | 50 | 14.2 | 1069 | 16.0 | 73 | 15.7 | 50 | 15.4S |
| Occupational category | | | | | | | | |
| Skilled Non-manual | 59 | 16.7 | 1419 | 21.3 | 85 | 18.3 | 57 | 17.6 |
| Skilled Manual | 58 | 16.4 | 1079 | 16.2 | 76 | 16.4 | 57 | 17.6 |
| Unskilled Non-manual | 148 | 41.9 | 3062 | 45.9 | 216 | 46.6 | 147 | 45.4 |
| Unskilled Manual | 88 | 24.9 | 1105 | 16.6 | 87 | 18.8 | 63 | 19.4 |
| Working time | | | | | | | | |
| Full-time | 261 | 73.9 | 4811 | 72.2 | 361 | 77.8 | 232 | 71.6 |
| Part-time | 92 | 26.1 | 1854 | 27.8 | 103 | 22.2 | 92 | 28.4 |

consideration is the research question being asked. GBTM is useful for research questions concerning taxometric or diagnostic issues, rather than questions concerning distinct subsamples. For example, GBTM could be used to identify groups of people who report clinically significant change during treatment. Individuals who respond to treatment are not conceptualised as a different population of individuals from those who do not. Instead, GBTM is used to pragmatically subcategorise a continuum of responses for closer examination. If prior theory and research suggest that distinct latent subpopulations exist, latent growth mixture modelling (LGMM) and, in particular, latent class growth analysis (LCGA) may be a better conceptual choice (although these situations are likely rare) [31].

Our outcome was sickness absence, measured as the number of days per quarter in a continuous variable, showing that the differences in outcomes between the two used software were small. However, it is possible that we might have found other results if we had used another outcome. For example, if the outcome was rarer, or dichotomous, it might be that the trajectories produced with different software are more or less similar. The size of the data is also assumed to play a notable role. Any trajectory analyses is affected by the size of the data and the type of outcome. However, for a continuous outcome, it is probable that the results do apply across datasets.

As far as we know, there is no clear evidence that any of the available software are superior to others [32]. Our results confirm this settlement. The optimal number of trajectories was not clear using one software or another, growth factors differ according to the software, individuals were differently distributed in trajectories (Up & Down: 5.9% in MPlus, 6.6% in SAS; Low Stable: 75.3% in Mplus, 76.0% in SAS; Increasing: 10.1% in Mplus, 10.7% in SAS; Decreasing: 8.7% in Mplus, 6.6% in SAS), and also there were differences in the start values of the trajectories (the Mplus trajectories tended to start slightly higher than the SAS trajectories). However, the magnitude of these differences was small, and the distribution of individuals between the trajectory groups by respective software were within a few percentage points of each other. We have not made any statistical tests of the differences between our two analyses; that is not feasible given that they were run in different software. Instead we have presented the results, and evaluated whether the trajectories produced appear similar in their numbers and shape. The conclusions drawn regarding the association between trajectory group membership and the covariates are the same regardless of software used for the analysis (Table 3).

Therefore, these models, even though they are powerful and useful to use for longitudinal data, contain many unknowns and limitations that may hinder their application and interpretation. For example, the selection criteria regarding the optimal number of trajectory groups are somewhat arbitrary, and it is up to each researcher to balance statistical criteria vs. meaningful interpretations. There are several model fit indices that need to be balanced, both related to overall model fit and related to individual trajectories and group assignment [22, 27, 28, 33]. Like previous researchers, we recommend producing spaghetti plots to compare the actual trajectories with the estimated trajectory [33]. Furthermore, the method may not always be comparable across different heterogeneous groups. This means that comparability across different study populations may be challenging–as is often the case in studies, also when using other analytical methods. The identified trajectories are approximations of the actual course of development and thus should be interpreted as statistically significant simplifications. Furthermore, a reasonably large sample size is required as it with a small sample size is difficult to identify distinct trajectories or meaningfully study determinants if latent groups are small. Most likely, researchers know more about the data than the software, so preliminary plots and descriptive statistics are very useful and highly recommended before deciding any software or starting running trajectory models.

Further studies are needed to go deeper into methodological aspects around these sophisticated statistical techniques. This will make research easier for epidemiologists and other researchers who are dealing with longitudinal study designs.

## Author Contributions

**Conceptualization:** Laura Serra, Kristin Farrants, Kristina Alexanderson, Mónica Ubalde, Tea Lallukka.

**Data curation:** Laura Serra, Mónica Ubalde.

**Formal analysis:** Laura Serra, Kristin Farrants.

**Methodology:** Laura Serra, Kristin Farrants, Mónica Ubalde, Tea Lallukka.

**Software:** Laura Serra, Kristin Farrants.

**Supervision:** Kristina Alexanderson, Tea Lallukka.

**Writing – original draft:** Laura Serra.

**Writing – review & editing:** Laura Serra, Kristin Farrants, Kristina Alexanderson, Mónica Ubalde, Tea Lallukka.

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
