## [Decision Letter · Decision Letter 0]

27 Apr 2021

PONE-D-20-30783

Trajectory analyses in insurance medicine studies: examples and key methodological aspects and pitfalls

PLOS ONE

Dear Dr. Serra,

Thank you for submitting your manuscript to PLOS ONE. After careful consideration, we feel that it has merit but does not fully meet PLOS ONE’s publication criteria as it currently stands. Therefore, we invite you to submit a revised version of the manuscript that addresses the points raised during the review process.

The reviewers raised a number of concerns about the methodology, the statistical analysis, and the presentation of the outcomes. The reviewers' comments can be viewed in full, below.

We look forward to receiving your revised manuscript.

Kind regards,

Natasha McDonald, PhD

Associate Editor

PLOS ONE

Journal Requirements:

3. Thank you for stating the following before the Acknowledgments Section of your manuscript:

'Funding

This work was supported by the Academy of Finland (Grants #287488 and #319200) and by the Swedish

Research Council for Health, Work and Welfare (grant number 2017-01943).'

'The author(s) received no specific funding for this work.'

Reviewers' comments:

Reviewer's Responses to Questions

**Comments to the Author**

1. Is the manuscript technically sound, and do the data support the conclusions?

Reviewer #1: Yes

Reviewer #2: Yes

Reviewer #3: Partly

2. Has the statistical analysis been performed appropriately and rigorously? 

Reviewer #1: Yes

Reviewer #2: Yes

Reviewer #3: No

3. Have the authors made all data underlying the findings in their manuscript fully available?

Reviewer #1: No

Reviewer #2: Yes

Reviewer #3: Yes

4. Is the manuscript presented in an intelligible fashion and written in standard English?

Reviewer #1: No

Reviewer #2: Yes

Reviewer #3: Yes

5. Review Comments to the Author

Reviewer #1: This paper describes an example of the use of trajectory analyses in both SAS and Mplus. The differences between the results are highlighted and important considerations when conducting the analyses are given.

The data has not been made available but it is explained that the data are restricted and available to those with permission only.

The manuscript has some grammatical issues and would benefit from additional feedback to ensure all sentences are structured properly. I have identified a couple of example sentences that need reworking below. I have also noted some other areas for improvement.

P3 line 7: Reword sentence beginning ‘Administrative registers..’.

P3 line 5 from bottom: It is stated that conditional independence is assumed in GBTM models. However, the use of the term GBTM varies and in some cases includes Growth Mixture Modelling which does not assume conditional independence. Please provide a reference for the definition of GBTM that is being used or specify your own definition.

P8 paragraph 1: The method of fitting a multinomial logistic regression for the association between covariates and trajectory group membership, following the estimation of the trajectory groups (3 step method), is not remedied by including the descriptive statistics. The covariate estimates will be biased due to the uncertainty in the group assignments. Vermunt’s improved 3 step approach (Vermunt 2010) or simultaneous estimation of the covariate effects with the trajectory model should be used to avoid this bias. (Davies, Giles & Glonek 2018)

P8 paragraph 2: Reword second sentence.

P5 last paragraph: The results for Mplus and SAS in terms of the number of groups selected should be presented in the Results section rather than the Methods.

Davies CE, Giles LC and Glonek GFV. Performance of methods for estimating the effect of covariates on group membership probabilities in group-based trajectory models. Statistical Methods in Medical Research 2018; 27(10): 2918-2932.

Vermunt JK. Latent class modeling with covariates: two improved three-step approaches. Political Analysis 2010; 18: 450–469.

Reviewer #2: In large population-level longitudinal studies, trajectory-based analyses provide an alternative approach that can be very useful in summarizing long-term behaviors/characteristics which are dynamic in nature.

Recently this method for identifying trajectories/patterns has increasingly been applied to various medical, sociological and public health research. However, there are still many questions and concerns associated with the use of this technique that need consideration, as stated by Serra et al. in this paper as well.

This well-written paper provides a detailed guide for researchers considering the application of trajectory-based modeling methods in their analysis, and this will be a very timely and much-needed addition to the literature.

Major Revisions: No major revisions.

Minor Revisions:

- Introduction – Typos in line 6 “In recent years, improvements in data collections and have allowed”, change to “collection” and remove “and”.

- Figure 1 – Please include labels to clearly identify that the left graph is from MPlus and right one from SAS.

Reviewer #3: Major comments:

- The authors focus on one type of outcome (e.g., sickness absence) to generalize to other types of outcomes. It would be helpful context to describe whether the authors expect their results to truly generalize to other outcomes; I suspect it may not given other types of data, ways of classifying outcomes, and commonness of outcomes.

- The underlying methods are different, so it is not clear that they can be directly comparable. The comparisons across software are interesting, but perhaps less useful given that there do not appear to be massive differences within method across software (and the authors do not provide enough information to be truly sure that they are making appropriate comparisons). It is also not clear from the methods whether the results are “meaningful” without using CIs or other types of methods to be sure that they are truly different.

o The (lack of) differences across software within method should be better illustrated in the abstract.

Minor comments:

- Title: “insurance medicine studies” is a bit awkward; recommend “insurance claims data” or similar

- Prior work: The authors should reference this study in the discussion (https://onlinelibrary.wiley.com/doi/abs/10.1002/pds.4917), given that it covers similar questions related to model fit, number of trajectory groups, and interpretation.

- Abstract: The authors describe “substantial differences” in the Abstract without providing any quantitative evidence to support their conclusions. For the casual reader, there is not enough information.

- The authors should describe better (Page 3) in introduction how GBTM relate to LCGA and GMM methods.

6. PLOS authors have the option to publish the peer review history of their article (what does this mean?). If published, this will include your full peer review and any attached files.

Reviewer #1: No

Reviewer #2: **Yes: **Mufaddal Mahesri

Reviewer #3: No

---

## [Author Response · Author response to Decision Letter 0]

16 Aug 2021

Editorial Office, the Plos One

Dear Editor,

Thank you for the comments on our study “Trajectory analyses in insurance medicine studies: examples and key methodological aspects and pitfalls”. We were very happy to learn that our manuscript was found to have merit, and that it could be suitable for publication after a revision.

We have now carefully revised the manuscript, considering all the points raised by the reviewers regarding about the methodology, the statistical analysis, and the presentation of the outcomes. Please find our point by point rebuttal letter attached in the submission. We have highlighted the changes in the manuscript using the track changes mode in MS Word (document labeled 'Revised Manuscript with Track Changes'). As requested, we have also downloaded an unmarked version of our revised paper without tracked changes (document labelled ‘Manuscript’). We have further ensured that our manuscript meets also all the other PLOS ONE's style requirements.

Finally, as requested, we have revised our statement regarding data sharing. Please be advised that due to legal restrictions regarding sharing of sensitive data at individual level, we cannot share these data. The datasets supporting the findings of this study are owned by a third-party organization. In particular, they are based on registers from the Spanish Social Security and the Catalan Institute for Medical and Health Evaluations. A record linkage agreement protocol between both institutions and the Centre for Research in Occupational Health ensures the confidentiality of the databases, which are anonymized to the authors and are not publicly available.

We think that this revised manuscript is now suitable for publication in your journal and look forward to hearing from you.

Sincerely yours,

Laura Serra Saurina

Center for Research in Occupational Health (CiSAL), Department of Experimental and Health

Sciences, Pompeu Fabra University, Barcelona, Spain.

---

## [Decision Letter · Decision Letter 1]

28 Jan 2022

Trajectory analyses in insurance medicine studies: examples and key methodological aspects and pitfalls

PONE-D-20-30783R1

Dear Dr. Serra,

We’re pleased to inform you that your manuscript has been judged scientifically suitable for publication and will be formally accepted for publication once it meets all outstanding technical requirements.

Kind regards,

James Mockridge

Staff Editor

PLOS ONE

Additional Editor Comments (optional):

Reviewers' comments:

Reviewer's Responses to Questions

**Comments to the Author**

1. If the authors have adequately addressed your comments raised in a previous round of review and you feel that this manuscript is now acceptable for publication, you may indicate that here to bypass the “Comments to the Author” section, enter your conflict of interest statement in the “Confidential to Editor” section, and submit your "Accept" recommendation.

Reviewer #1: All comments have been addressed

Reviewer #2: All comments have been addressed

Reviewer #3: All comments have been addressed

2. Is the manuscript technically sound, and do the data support the conclusions?

Reviewer #1: (No Response)

Reviewer #2: Yes

Reviewer #3: Yes

3. Has the statistical analysis been performed appropriately and rigorously? 

Reviewer #1: (No Response)

Reviewer #2: Yes

Reviewer #3: Yes

4. Have the authors made all data underlying the findings in their manuscript fully available?

Reviewer #1: (No Response)

Reviewer #2: Yes

Reviewer #3: Yes

5. Is the manuscript presented in an intelligible fashion and written in standard English?

Reviewer #1: (No Response)

Reviewer #2: Yes

Reviewer #3: Yes

6. Review Comments to the Author

Reviewer #1: (No Response)

Reviewer #2: (No Response)

Reviewer #3: (No Response)

7. PLOS authors have the option to publish the peer review history of their article (what does this mean?). If published, this will include your full peer review and any attached files.

Reviewer #1: No

Reviewer #2: **Yes: **Mufaddal Mahesri

Reviewer #3: No

---

## [Editor Report · Acceptance letter]

3 Feb 2022

PONE-D-20-30783R1 

Trajectory analyses in insurance medicine studies: examples and key methodological aspects and pitfalls 

Dear Dr. Serra:

I'm pleased to inform you that your manuscript has been deemed suitable for publication in PLOS ONE. Congratulations! Your manuscript is now with our production department. 

Kind regards, 

on behalf of

Dr James Mockridge 

Staff Editor

PLOS ONE